# Intravenous Immunoglobulins as Immunomodulators in Autoimmune Diseases and Reproductive Medicine

**DOI:** 10.3390/antib12010020

**Published:** 2023-03-02

**Authors:** Tsvetelina Velikova, Metodija Sekulovski, Simona Bogdanova, Georgi Vasilev, Monika Peshevska-Sekulovska, Dimitrina Miteva, Tsvetoslav Georgiev

**Affiliations:** 1Medical Faculty, Sofia University St. Kliment Ohridski, 1 Kozyak Str., 1407 Sofia, Bulgaria; 2Department of Anesthesiology and Intensive Care, University Hospital Lozenetz, 1 Kozyak Str., 1407 Sofia, Bulgaria; 3First Department of Internal Medicine, Medical Faculty, Medical University of Varna, 9000 Varna, Bulgaria; 4Clinic of Neurology, UMHAT “Sv. Georgi”, Faculty of Medicine, Medical University of Plovdiv, 4000 Plovdiv, Bulgaria; 5Department of Gastroenterology, University Hospital Lozenetz, 1 Kozyak Str., 1407 Sofia, Bulgaria; 6Department of Genetics, Faculty of Biology, Sofia University St. Kliment Ohridski, 8 Dragan Tzankov Str., 1164 Sofia, Bulgaria

**Keywords:** intravenous immunoglobulins (IVIGs), autoimmune diseases, reproductive failure, immunomodulation, Kawasaki, SLE, myositis, anti-phospholipid syndrome, Guillen–Barre syndrome

## Abstract

Intravenous administration of immunoglobulins has been routinely used for more than 60 years in clinical practice, developed initially as replacement therapy in immunodeficiency disorders. Today, the use of intravenous immunoglobulins (IVIGs) is embedded in the modern algorithms for the management of a few diseases, while in most cases, their application is off-label and thus different from their registered therapeutic indications according to the summary of product characteristics. In this review, we present the state-of-the-art use of IVIGs in various autoimmune conditions and immune-mediated disorders associated with reproductive failure, as approved therapy, based on indications or off-label. IVIGs are often an alternative to other treatments, and the administration of IVIGs continues to expand as data accumulate. Additionally, new insights into the pathophysiology of immune-mediated disorders have been gained. Therefore, the need for immunomodulation has increased, where IVIG therapy represents an option for stimulating, inhibiting and regulating various immune processes.

## 1. Introduction

Immunoglobulins are proteins (antibodies) produced by plasma cells that mainly identify and neutralize foreign bodies, such as microbial agents, bacteria, viruses, fungi or cancer cells [1]. Intravenous administration of immunoglobulins has been routinely used for more than 60 years in clinical practice. They were developed in the 1960s and were initially used as replacement therapy in immunodeficiency disorders [2]. Today, intravenous immunoglobulins (IVIGs) are embedded into modern algorithms for managing a few diseases. However, in most cases, their application is off-label and thus different from their registered therapeutic indications according to the summary of product characteristics.

IVIGs belong to plasma products prepared from the serum of several thousand healthy donors per batch since the large numbers of donors increase the number of individual antibodies [3]. IVIG products contain a high titer of antibodies against specific antigens and are used to regulate the immune reactions in patients with disorders of the immune system. The majority of commercial preparations of IVIGs consist primarily of polyclonal immunoglobulin G (IgG) (>90%) [4]. In addition, other immunoglobulins, such as IgM, IgA and soluble molecules (i.e., human leukocyte antigen, HLA), are present in small amounts [5].

Initially, immunoglobulins were administered by intramuscular injection; then, intravenous immunoglobulins were also introduced. Three generations of immunoglobulin preparations for intravenous administration are known: preparations obtained with the participation of enzymes, preparations containing chemically modified immunoglobulins and preparations in which immunoglobulins are processed at low pH [6].

In this review, we present the state-of-the-art use of IVIGs in various autoimmune conditions and immune-mediated disorders associated with reproductive failure, as approved therapy, based on indications or off-label. IVIGs are often an alternative to other treatments, and the administration of IVIGs continues to expand as data accumulate. Additionally, new insights into the pathophysiology of immune-mediated disorders have been gained. Therefore, the need for immunomodulation has increased, where the IVIGs represent an option for stimulating, inhibiting and regulating various immune processes.

## 2. Immune Mechanisms of IVIGs as Immunomodulators

The immunoregulatory effects of IVIGs in autoimmune and inflammatory diseases depend on various mechanisms, including the interaction of immunoglobulin Fc portions with Fc receptors on lymphocyte repertoires through variable regions of infused immunoglobulins [7]. In addition, IVIGs modulate B and T lymphocyte activation and effector functions, neutralize pathogenic autoantibodies, interfere with antigen presentation and usually possess a robust anti-inflammatory effect (via interactions with cytokines, chemokines, complement system components, endothelial cells, etc.) [8]. IVIGs’ immunomodulatory potential in patients with various immune-mediated, inflammatory and autoimmune diseased results from a number of complex mechanisms working together (Figure 1).

In contrast to the well-known use of IVIGs as replacement treatment in primary immune deficiencies (i.e., antibody deficiency), when administered as immunomodulators, IVIGs influence more than one immune mechanism, with many innate and adaptive immune pathways being targeted. In addition, many distinct but not mutually exclusive immunological effects have been demonstrated [9]. Therefore, it is difficult to determine a common mechanistic understanding of the IVIG mode of action. Simultaneously, upon administration, IVIGs modulate and regulate the functions of various immune cells and molecules. 

### 2.1. IVIG Effects on Antigen-Presenting Cells and Proinflammatory Cytokine Production

Nevertheless, data so far agree on IVIG regulatory properties, such as reducing the production of proinflammatory cytokines (e.g., tumor necrosis factor α (TNF-α), interleukin-(IL)1α, IL-6), down-regulating adhesion molecule, chemokine and chemokine-receptor expression and neutralizing superantigens [9]. IVIGs inhibit the activation of macrophages and monocytes by affecting the transcription of inflammatory genes and reducing the circulating levels of proinflammatory cytokines [10]. IVIGs affect dendritic cells’` maturation and differentiation via inhibition or stimulation, depending on the doses administered.

### 2.2. IVIG Effect on NK and NKT Cells

IVIGs may reduce both the number and functional activity of NK and NKT cells [11,12]

However, IVIGs can enhance NK cells’ anti-tumor activity and antibody-dependent cell-mediated cytotoxicity in peripheral blood by regulating the IL-12 production of monocytes [13,14]. 

### 2.3. IVIG Effects on Adaptive Immune Cells

IVIGs may inhibit the expansion of autoreactive B cells, thus controlling the production of autoantibodies by inducing G1 phase arrest and apoptosis in B cells [15,16]. 

Regarding T cells, IVIGs suppress their proliferation and cytokine production due to direct cell interaction or the suppression of IL-2 while increasing CD4^+^ CD25^+^ Foxp3^+^ Tregs T regulatory cells and inhibiting Th17 and associated cytokines IL-17A, IL-17F, IL-21 and CCL20 [10,16,17,18,19]. 

### 2.4. Dose-Dependent Effects of IVIGs and Relevant Immunoglobulin Receptors

IVIG administration can exert proinflammatory actions in some circumstances, depending on the interaction and doses [20]. Usually, high-dose administration of IVIGs elicits anti-inflammatory effects. At low doses, IVIGs activate complement or innate immune effector cells via binding their receptors (FcγR) to the Fc (crystallizable fragment) of IVIGs and exerting more pro-inflammatory effects. Fcγ receptors and their relative expression and affinities may also regulate the overall effect of IVIGs, establishing a threshold for activation of immune effector cells. However, this balance between IVIG pro- and anti-inflammatory effects can be altered by many factors, such as different cytokines, pro- and anti-inflammatory stimuli, phagocytosis, degranulation, antigen presentation and antibody-dependent cell cytotoxicity [20].

On the other hand, mechanisms that involve the IgG antigen-binding fragment (Fab) are linked to anti-inflammatory or immunomodulatory activities [21]. Since IVIG preparations contain many antibodies with distinct specificities, some therapeutic effects are expected to rely on Fab binding to various antigens (i.e., proteins, cell-surface receptors). This well-established mechanism involves Fab-dependent interaction and is the idiotypic–anti-idiotypic network. An array of anti-idiotypic immunoglobulins is assumed to target B cells expressing these idiotypes. Therefore, the IVIG administration may lead to the down-regulation or elimination of autoreactive clones [21].

Gelfand et al. summarized the different activities of IVIGs, such as Fab-mediated activities (suppression or neutralization of autoantibodies, cytokines, activated complement components; restoration of idiotypic–anti-idiotypic networks, blockade of leukocyte adhesion molecule binding and specific immune cell–surface receptors; modulation of maturation and function of dendritic cells) and Fc-dependent activities (i.e., blockade of the neonatal FcRn and activating FcγR, up-regulation of inhibitory FcγRIIB, immunomodulation by sialylated IgG) [7].

Some of the effects of IVIGs are connected with binding the potentially harmful complement fragments, blocking them and preventing the deposition of immune complexes and subsequent damage of the target organ via destruction or aggravated inflammation [22].

Additionally, IVIGs may improve the response to glucocorticosteroids (GCs) by improving glucocorticosteroid-receptor binding, especially in patients with severe, glucocorticosteroid-resistant conditions (i.e., severe asthma) [23]. The pleiotropic effect of IVIGs has been studied, and still, no single mechanism can explain all of their effects. Moreover, IVIG effects extend beyond their half-life, suggesting more than just interference or clearance of the pathological autoantibodies (Figure 1).

## 3. IVIG Treatment for Autoimmune Diseases

IVIG treatment was introduced for immunodeficient patients for replacement therapy with a dosage regimen of 0.2–0.4 g/kg body weight. The treating doses for patients with immune-mediated diseases are usually higher—1–2 g/kg body weight. After replacement therapy, the expected blood levels of IgG vary between 12–14 mg/mL, whereas after high-dose treatment, the anticipated blood levels of IgG are 25–35 mg/mL [24].

Prior to IVIG administration, serum immunoglobulin levels must be measured. This is recommended because patients with selective IgA deficiency may develop an anaphylactic reaction upon receiving IVIGs due to existing anti-IgA antibodies in their serum. Additionally, a pre-existing hyperglobulinaemia may aggravate, leading to a hyperviscosity state [25].

When we discuss using IVIGs as immunomodulators, double-blind, placebo-controlled studies have been conducted for different indications to establish the efficacy and safety of use. However, a limited number of controlled trials (typically with a single product) have been carried out for some conditions. Moreover, only a few studies compared different products and brands [7]. Therefore, IVIGs have often been used for off-label indications in many countries. Non-medically-based IVIG use should be avoided when there is not enough evidence, for example, to treat autism and chronic fatigue [7].

Among the FDA-approved indications are primary immunodeficiencies, chronic lymphocytic leukemia, pediatric HIV infection, Kawasaki’s disease, allogeneic bone marrow transplantation, chronic inflammatory demyelinating polyneuropathy, kidney transplantation involving a recipient with a high antibody titer or an ABO-incompatible donor and multifocal motor neuropathy [7]. 

Additional approved conditions, if the needed criteria are met, are neuromuscular disorders (i.e., Guillain–Barré syndrome, relapsing–remitting multiple sclerosis, myasthenia gravis, refractory polymyositis, polyradiculoneuropathy, Lambert–Eaton myasthenic syndrome, opsoclonus–myoclonus, Birdshot retinopathy and refractory dermatomyositis), rheumatic diseases (i.e., ANCA-positive systemic vasculitis, polymyositis, dermatomyositis, anti-phospholipid syndrome, rheumatoid arthritis (RA) and Felty’s syndrome, systemic lupus erythematosus (SLE), juvenile idiopathic arthritis (JIA)), hematologic disorders (i.e., autoimmune hemolytic anemia, severe anemia associated with parvovirus B19, autoimmune neutropenia, neonatal alloimmune thrombocytopenia, HIV-associated thrombocytopenia, graft-versus-host disease, CMV infection or interstitial pneumonia after bone marrow transplantation), dermatologic disorders (i.e., pemphigus vulgaris, pemphigus foliaceous, bullous pemphigoid, mucous–membrane (cicatricial) pemphigoid, epidermolysis bullosa acquisita, toxic epidermal necrolysis or Stevens–Johnson syndrome, necrotizing fasciitis), recurrent spontaneous abortions and sepsis. [7,9]. 

For some disorders, such as RA, IVIGs may be useful in subsets of RA patients where anti-cytokine blockers or rituximab are contraindicated. Patients with RA and concomitant vasculitis, overlap “rhupus” syndrome, severe active infections and pregnancy are examples of such subgroups of patients. IVIGs may also be used to treat juvenile chronic arthritis (JCA), and adult Still’s disease [26]. However, other sources do not support the use of IVIGs for RA [27].

There is solid evidence that under certain conditions and a specific dosage regimen, IVIGs can play the role of immunomodulatory and/or even life-saving therapy to modify the course of the underlying disease in patients with autoimmune inflammatory myositis, systemic lupus erythematosus and catastrophic anti-phospholipid syndrome [28]. There are also case reports and series for the effective use of IVIGs in progressive systemic sclerosis, ANCA-associated and large vessel vasculitis. Their use could also serve as corticosteroid-sparing therapy, i.e., with a tendency to reduce the therapeutic or maintenance dose of GCs. We will cover the use of IVIGs for these conditions in the following sections.

### 3.1. Idiopathic Autoimmune Inflammatory Myositis

Idiopathic inflammatory myopathies (IIM) are diseases that involve the skeletal muscles but can also affect other internal organs such as the gastrointestinal tract, cardiovascular system, lungs and skin [29]. The diseases included in this group are polymyositis, dermatomyositis, inclusion body myositis (IBM), overlap-myositis, immune-mediated necrotizing myositis and antisynthetase syndrome, according to the current classification [30]. Although their clinical presentation is similar, there are differences in the pathogenesis of different types of IIM. For example, in polymyositis and IBM, sensitized CD8+ cytotoxic T cells [31] recognize previously unidentified muscle antigens, leading to phagocytosis and the necrosis of fibers [32]. In dermatomyositis, where a characteristic skin rash is observed, intramuscular microangiopathy occurs, mediated by the attacking complement membranolytic complex C5b-9 [33]. This results in capillary loss, muscle ischemia, muscle fiber necrosis and perifascicular atrophy.

The “conventional” therapy for IIM includes high doses of GCs and immunosuppressive drugs such as cyclophosphamide, cyclosporine A, methotrexate, mycophenolate mofetil, and azathioprine. In refractory cases, IVIGs are also used in treating inflammatory myositis, both polymyositis and dermatomyositis [34]. The effect of IVIG administration is immunomodulatory rather than immunosuppressive [35]. However, the exact mechanisms of action are not fully understood. As a result, their intravenous use leads to a decrease in the migration of inflammatory cells in the muscle fibers, a reduction in the expression of TGFβ in the muscles, inhibition of the maturation of dendritic cells and B-cell proliferation, activation of regulatory T cells (Treg cells) and modulation of proinflammatory cytokines [36].

The recommended dosage of IVIGs for IIM is 2 g/kg, usually divided into two to five separate daily doses with a therapeutic course of 3 to 6 months. IVIG therapy is not usually used as first-line therapy in IIM. Instead, it is often used in refractory, exacerbating, rapidly progressive, or severe polymyositis/dermatomyositis or in patients with contraindications to high-dose GCs. At this stage, no precise guidelines/recommendations have been adopted for IVIG infusions and added to the standard immunosuppressive therapy. However, there is evidence that in patients with dermatomyositis that is refractory to standard treatment, IVIGs in combination with corticosteroids significantly improve muscle strength and motor function and reduce serum creatine phosphokinase (CPK) compared to a placebo [37]. The effect is most pronounced in patients with esophageal involvement [38] or pulmonary involvement [39], as well as in elderly patients [40]. A similar effect was observed in patients with IBM, lasting 2 to 4 months after the administration of IVIGs [41].

Sufficient evidence demonstrated that the administration of immunoglobulins (intravenous or subcutaneous) prolongs life in patients with inflammatory myopathies [42]. Regarding side effects, the medication is relatively well tolerated in patients with IIM. The most common adverse drug reactions observed are headache, fever and nausea [43]. Particular attention is paid to the possibility of thromboembolic incidents.

The ProDERM trial was the first to evaluate the long-term effectiveness and safety of IVIGs (Octagam 10%) in dermatomyositis in a placebo-controlled, blinded, randomized trial. Patients in this trial were given high-dose IVIGs (2.0 g/kg) for up to 40 weeks. However, following an FDA suggestion, the investigators could reduce the dosage to 1.0 g/kg starting at week 28 if the patients’ condition permitted. Because this trial used long-term IVIG medication at a potentially high dosage and because patients with dermatomyositis are at a greater risk of thromboembolic events and hemolytic transfusion responses, special attention was paid to monitoring these complications [44].

At 16 weeks, 79% of IVIG patients vs. 44% of placebo patients had a total improvement score of at least 20. Over the course of 40 weeks, the IVIG-treated group experienced 282 treatment-related side effects, including headache (42% of patients), pyrexia (19%) and nausea (16%). In addition, nine significant adverse events were thought to be connected to IVIGs, including six thromboembolic events [45].

Kocoloski et al. reported that IVIG therapy was linked to considerable improvement for immune-mediated necrotizing myositis patients, with 85% of patients satisfying clinically meaningful response criteria. Furthermore, according to the ACR/EULAR 2016 myositis response criteria, most patients improved with IVIG treatment. The latter was also well-tolerated [46].

### 3.2. Systemic Lupus Erythematosus

Systemic lupus erythematosus (SLE) is an autoimmune disease with a heterogeneous clinical manifestation involving symptoms and syndromes from many organs and systems. It was not until the late 1980s that IVIGs were used to treat SLE [47]. The use of IVIGs in patients with SLE has several indications (i.e., pancytopenia, central nervous system (CNS) involvement, refractory thrombocytopenia, secondary anti-phospholipid syndrome and lupus nephritis).

It has been suggested that patients with SLE have a dysregulation of the FcγR system, where the balance between activating and inhibitory FcγR signaling is disturbed. Although the exact mode of action of IVIGs is not fully understood, it is suggested that the IgG Fc segments of IVIGs bind to macrophage Fc receptors, which in turn inhibit autoantibody binding to these receptors. Furthermore, IVIGs exert their therapeutic properties by inhibiting membrane attack complex formation by restraining the Fc segment from complement components C3b and C4b [48].

As we pointed out earlier, IVIG therapy leads to the suppression of T cells [49]. Furthermore, IVIGs decrease the Th1/Th2 ratio, which leads to a change in the peripheral Th1/Th2 balance in favor of the Th2 subpopulation [50]. In addition, IVIGs decrease the activation of FcRIIA and FcRIIC and/or increase the inhibitory FcRIIB. Ultimately, the therapy leads to the inhibition of complement-mediated injury, modulation of cytokines and cytokine antagonists production, T- and B-cells’ function, induction of apoptosis in lymphocytes and monocytes and reduction in the production and neutralization of pathological autoantibodies [51].

The use of IVIGs for treating SLE does not yet have official approval from the Food and Drug Administration (FDA); however, the drug is used off-label in cases where patients are refractory to standard therapy and/or have contraindications. The therapeutic dose of IVIGs in SLE is 2 g/kg divided into five daily doses of 400 mg/kg each to prevent the risk of adverse reactions [52]. Diseases such as severe congestive heart failure, renal failure or evidence of hypercoagulation are a contraindication for the therapy. Depending on the patient’s response and the objective signs of the disease, long-term therapy is carried out for a period of 6–12 months, and treatment courses are repeated every 4–6 weeks. For now, there is evidence that the administration of IVIGs can have a GCs-sparing effect, both on the maintenance dose and the cumulative GCs dose in patients with SLE. 

However, data regarding their effect on complement fractions and the reduction of lupus-specific antibody levels are conflicting. IVIGs are known to accelerate autoantibody catabolism by binding to a specific Fc receptor found on endothelial cells called FcRn [52]. FcRn is a transport receptor that binds intracellular IgG and protects it from catabolism and lysosomal degradation. Saturation of FcRn receptors by IVIG treatment prevents the binding of endogenous IgG autoantibodies, which accelerates their degradation and reduces levels of pathogenic autoantibodies.

On the other hand, a proven effect was observed on the reduction of proteinuria induced by lupus nephritis [53]. The Fc receptors suggested to contribute to the deposition of IgG in the kidney in SLE are FcγRI (activating receptor for monomeric Ig), FcγRII (inhibitory immune complex receptor) and FcγRIV (activating immune complex receptors). IVIGs can beneficially affect the balance between activating and inhibitory Fc receptors in the kidney, resulting in more significant degradation and urinary excretion of autoantibodies to minimize renal parenchymal damage [54]. IVIG therapy is indicated in patients with lupus nephritis who have contraindications for conducting conventional immunosuppressive treatment, do not respond to standard therapy, have a concomitant superimposed infection or during pregnancy [55]. 

IVIGs have also been shown to inhibit the expression of human leukocyte antigen and CD80/86 on dendritic cells leading to a reduction in the differentiation of dendritic cells from blood monocytes, which has an immunomodulatory effect [56]. In addition, clinical studies indicate that IVIG therapy reduces disease activity indices [57]. IVIGs also reduce proinflammatory cytokines such as TNF-α and IL-6 [58].

One of the most significant advantages of IVIG therapy is that, unlike conventional immunosuppressants, which predispose to systemic infections, IVIGs actually prevent infections and provide passive immunity [59]. In addition, the side effects of immunosuppressants, such as neocarcinogenesis, gonadotoxicity, hemorrhagic cystitis and cytopenias, are also avoided.

Additionally, clinical response to therapy in milder forms of SLE is seen as an improvement of thrombocytopenia, clinical improvement of facial erythema and arthritis [60], as well as myalgias and fever [61]. IVIG therapy rapidly increases platelet counts in autoimmune thrombocytopenia associated with SLE. However, it is recommended to limit therapy to patients with life-threatening thrombocytopenia who are refractory to oral GCs because of the high cost of treatment and the relatively short-lived response to treatment [62]. Cases have been described where IVIGs improved therapy-resistant cutaneous lupus [63].

Although there are no official indications in the therapeutic guidelines, IVIGs are administered as an off-label therapy in patients with neuropsychiatric SLE, mainly based on the results from case series studies. Cases of patients with CNS vasculitis have already been presented where a significant improvement in ischemic changes was observed [64]. Improvement of mononeuritis was also described by Jose et al. [65]. In addition, IVIGs were used to prevent relapses of optic nerve neuromyelitis [66]. A good response was also observed in acute demyelinating peripheral polyneuropathy [67] and aseptic meningitis in SLE [68].

A severe side effect of IVIG therapy is nephrotoxicity due to therapy-induced renal tubular necrosis [69], which can lead to renal failure.

### 3.3. Anti-Phospholipid Syndrome

Anti-phospholipid syndrome (APS) is characterized by the presence of anti-phospholipid antibodies (aPL) (i.e., lupus anticoagulant (LA), anticardiolipin (aCL), anti-2 glycoprotein-I (2GPI) antibodies, anti-annexin antibodies), venous and arterial thromboses and recurrent fetal losses. The current concept for treating thrombotic APS is heparin administration, followed by long-term anticoagulation. In contrast, for obstetric APS, the therapy is low-dose aspirin (LDA) plus preventive unfractionated or low-molecular-weight heparin (LMWH) [70].

Tenti et al. focused on the 35 articles, 14 case reports, 9 case series and 12 clinical trials (9 open-label, 3 randomized controlled) published on IVIGs for APS, with a total of 802 patients, 99% of them being women [70]. However, the evidence for IVIG therapy in nonpregnant APS patients is scarce. 

In a study, the patients with high-risk aPL profiles were administered at a dose of 0.4 g/kg/daily IVIG infusions, in addition to conventional therapy (anticoagulants or antiplatelets) for 3 months to obtain primary or secondary thromboprophylaxis. Then, the patients were administered a monthly infusion of 0.4 g/kg/day for 9 months. A 5-year follow-up demonstrated no thrombosis that was clinically or instrumentally proven [71].

However, no significant differences were observed in aPL levels before and after IVIG treatment at 6, 12 and 24 months [55]. Therefore, primary or secondary thrombosis prophylaxis is still controversial, and there is no adequate therapy. Nevertheless, adding IVIGs to conventional treatment as an immunomodulator is promising and encouraging [72]. Furthermore, IVIG administration could be beneficial in preventing recurrent thrombosis in APS patients who are refractory to conventional anticoagulant therapy [70].

Regarding catastrophic APS, some studies employed IVIGs, demonstrating the beneficial effects of immunoglobulins when combined with the standard therapy or biologics (i.e., rituximab) [73], especially when the patients are refractory to conventional anticoagulant therapy [70]. In rare cases, catastrophic APS may be refractory to high-dose IVIGs, then plasma exchange could be performed [74].

### 3.4. Systemic Sclerosis

Progressive systemic sclerosis (SSc) is a chronic autoimmune disease characterized by progressive skin fibrosis, obliteration of microvasculature and excessive extracellular matrix deposition. In addition, it leads to multisystem dysfunction [75]. The etiology and pathogenesis of this disease are still not fully understood.

Given the heterogeneous clinical manifestation involving symptoms and syndromes from many organs and systems, the therapeutic challenges to treating this disease are still the subject of extensive research [76]. In addition, the condition is relatively rare, with a variable course and possible severe complications. Several immunomodulatory agents are also used in the therapeutic arsenal of SSc. At this stage, no drug has been proven effective in the long-term control of the disease; thus, treatment has mainly remained symptomatic in recent years [77,78]. New therapies are currently being tested and may potentially alter the disease process and overall clinical outcome [79]. IVIG therapy in patients with SSc has been used since 2000 in various therapeutic doses and regimens [75,76,77,78,79,80].

At this stage, there is a lack of definitive guidelines on when and how to administer IVIG treatment. The usual dose is 1–2 g/kg body weight distributed over 2–5 consecutive days, with the recommendation of 3–4 courses per year. According to the literature data, single cases have been described in which therapy has benefited skin involvement, musculoskeletal symptoms [78] and symptoms of interstitial lung disease. In addition, cases of IVIG-treated SSc patients with active diffuse cutaneous scleroderma (dcSSc) refractory to standard immunosuppressive therapy have been described, with improvement in the modified Rodnan skin score (mRSS) [79]. The same authors also describe the preservation of lung function without deterioration of forced vital capacity (FVC) at follow-up and the improvement in joint function. Still, definitive evidence of delay and/or improvement of interstitial lung disease at this stage is lacking. A similar effect on skin symptoms was also observed in patients with rapidly progressive skin involvement, with no effect on the immunological activity of the disease, i.e., on the antibody titer [78]. In addition, IVIGs reduce systemic inflammation and acute phase indicators [81] and help reduce the daily dose of corticosteroids at the end of treatment.

In patients with musculoskeletal involvement, they lead to a reduction in muscle weakness and pain, a reduction in joint pain and a reduction in serum creatine phosphokinase (CPK) levels [82]. Benefits on gastrointestinal symptoms following courses of IVIGs have been described, resulting in a reduction in the frequency and severity of symptoms of gastro-oesophageal reflux disease. Improvement of motility disorders of the gastrointestinal tract, both in the neuropathic and myopathic stages, is carried out by influencing antibodies against muscarinic-3 receptors (M3-R) [83]. There is no evidence of the influence of IVIG therapy on the manifestations of peripheral vasospasm (Raynaud’s syndrome).

IVIG therapy’s most common side effects are flu-like symptoms [84], headache, facial flushing, malaise, chills, fever, vomiting, diarrhea, nausea, myalgia, back pain, fatigue, dyspnea and changes in blood pressure [85]. These manifestations are reversible with prior application of analgesics, non-steroidal anti-inflammatory drugs, antihistamines or intravenous GCs. Nevertheless, late adverse reactions can be severe and include acute renal failure, thromboembolic vascular events (myocardial infarctions, cerebrovascular events, deep vein thrombosis and pulmonary embolism), aseptic meningitis, neutropenia, autoimmune hemolytic anemia, skin reactions, arthritis and pseudo hyponatremia [86].

### 3.5. Kawasaki Disease

Kawasaki disease is a self-limited acute vasculitis that affects small and medium-sized vessels [87]. It is among the leading causes of pediatric-acquired heart disease in developed countries and the second most common type of childhood vasculitis after Henoch–Schönlein purpura [88]. Although the inflammatory process resolves spontaneously in most patients, up to 25% of untreated patients present coronary artery involvement [89], which is reduced to less than 5% in children treated with high-dose intravenous immunoglobulin [90] by a still unknown mechanism [91].

The dosage and time of administration in the disease course remain debatable. However, a recent meta-analysis has shown that IVIGs in the early stage of disease onset might be associated with an increased risk of treatment unresponsiveness. On another note, a timely and adequate IVIG dosage could be a protective factor against the development of coronary artery lesions [90]. The most often prescribed IVIG therapy is at 2 g/kg. Nevertheless, older adolescents with more significant body weights sometimes require higher IVIG dosages, which leads to additional challenges and costs. It is unclear if a 2 g/kg dose of IVIGs is necessary for older children with Kawasaki disease. A study found no significant difference in hospitalization length, but the medical expenses were considerably greater. The number of IVIG side effects was too minor to compare. Based on the fact that IVIGs are a costly medicine, the dosage must be carefully examined [92]. 

### 3.6. ANCA-Associated Vasculitides

Antineutrophil cytoplasmic antibody (ANCA)-associated vasculitides (AAVs) are granulomatosis with polyangiitis (GPA, Wegener’s granulomatosis), microscopic polyangiitis (MPA) and eosinophilic granulomatosis with polyangiitis (EGPA, Churg–Strauss syndrome). The European League Against Rheumatism (EULAR) revised their recommendations for the treatment of AAV in 2016, including IVIGs [25].

A randomized, placebo-controlled trial investigated the potential of IVIGs for patients with AAV with a single course of a total dose of 2 g/kg in previously-treated AAVs with persistent disease activity. It was shown that IVIGs exerted less toxicity than conventional immunosuppressive agents while reducing disease activity. However, this effect was not maintained after 3 months [93]. 

Fortin et al. aimed to investigate the IVIGs as adjuvant therapy for WG as a therapeutic advantage over and above treatment with systemic corticosteroids in combination with immunosuppressants. One randomized controlled trial was included in the analysis. The decreased disease activity score was slightly more excellent for the IVIG treatment than the placebo, and the total adverse effects were fewer in the IVIG-treated group. However, the analysis could not confirm the therapeutic advantages of IVIGs above other conventional therapy, and the authors concluded that given the high cost of IVIGs, IVIGs should be limited to WG treatment in the context of well-conducted randomized controlled trials [94].

A study by Muso et al. also demonstrated a high safety profile of IVIGs at 0.4 g/kg/day administered for 5 consecutive days before or with conventional immunosuppressive therapy to 30 myeloperoxidase (MPO) ANCA-positive rapidly progressive glomerulonephritis patients [95]. 

Based on these studies, EULAR recommends adjunctive therapy with IVIGs for patients who fail to achieve remission and have a persistent low activity to help maintain remission [25]. In addition, ACR recommends the following: for GPA/MPA that is refractory to remission induction therapy, adding IVIGs (2 mg/kg as adjunctive therapy for short-term control, while waiting for remission induction therapy (i.e., cyclophosphamide or rituximab) to become effective. Additionally, according to ACR recommendations, IVIGs should not be used routinely to treat GPA/MPA [96]. 

Nevertheless, in the rare cases when patients with active disease cannot be treated with conventional immunosuppressive therapy (e.g., sepsis or pregnancy), IVIGs can be used as a short-term administration to allow for conventional remission induction therapies to take effect [97].

### 3.7. Gastrointestinal Autoimmune Diseases

An organ manifestation of autoimmune dysautonomia, autoimmune gastrointestinal dysmotility (AGID), is a newly characterized clinical condition that can be either an idiopathic or paraneoplastic phenomenon [98]. Generalized dysautonomia may be accompanied by gastrointestinal hypomotility or hypermotility, or it may be a feature of a multifocal paraneoplastic autoimmune neurological illness. The symptoms may include gastroparesis, colonic inertia or intestinal pseudoobstruction [99]. In a few rare cases, pyloric obstruction or anal spasms have also been reported as well. Early satiety, nausea, vomiting, bloating, diarrhea, constipation and involuntary weight loss are among the symptoms [100]. 

As far as treatment is concerned, there have been several options, among which is IVIG administration. In their study, Schofield et al., presented approximately 85% clinical improvement in IVIG-treated patients because of autoimmune dysautonomia. They included 38 patients, 8 of whom had GI dysmotility [101]. Kawanishi et al. reported another interesting case report about a 37-year-old woman who had been diagnosed with idiopathic chronic intestinal pseudoobstruction as a clinical presentation of AGID. They treated her with total parenteral nutrition, a gastrointestinal prokinetic agent and opiates as pain relievers. However, breakthrough pain continued; thus, Kawanishi et al. applied IVIGs with slight improvement [102]. AGID could also be a post-viral complication except for paraneoplastic and idiopathic characteristics, for example, the case reported by Montalvo et al. of a patient with AGID resulting from SARS-CoV-2. Despite various medications, her condition has worsened to total parenteral feeding. Hence, the IVIG administration has been initiated. The patient started to improve after the second infusion and tolerated oral nutrition. After four months of IVIG treatment, her symptoms significantly improved, and she tolerated a full oral diet without any symptoms [103]. 

In conclusion, IVIGs are safe and helpful in a minority of individuals with autonomic problems and evidence of autoimmunity, according to a growing body of research. In patients with severe illness who are unresponsive to pharmaceutical and lifestyle treatments, a 4-month IVIG trial should be considered.

### 3.8. Autoimmune Neurological Disorders

Moralez-Ruiz et al., in their systematic review and meta-analysis, focused on the efficacy of IVIGs in autoimmune neurological diseases, including Guillain–Barré syndrome, myasthenia gravis, chronic inflammatory demyelinating polyneuropathy, optic neuritis and multiple sclerosis [104]. The results demonstrated that IVIG administration outweighed the placebo, had similar efficacy as plasmapheresis and did not differ significantly from GCs [104]. 

Guillain–Barré syndrome (GBS) is an autoimmune-mediated disorder of the peripheral nervous system that is the most common cause of acute-onset flaccid paralysis in the developed world nowadays. It typically presents with weakness and sensory phenomena affecting the distal areas of the lower limbs at first and then ascending proximally, but several variants of the disease exist [105]. 

As the natural history of the disease usually follows a viral or bacterial infection in the previous few weeks, it is firmly believed that the pathogenesis of GBS includes an antibody response to microbial structures, especially those of Campylobacter jejuni, mimicking neuronal gangliosides and glycolipids [106]. In the most severe forms of the disease, where marked axonal degeneration heralds a grave prognosis, IgG antibodies against GM1, GD1b and/or GD1a gangliosides of peripheral neurons are encountered [107]. 

The first randomized controlled trial comparing the effectiveness of IVIGs vs. plasma exchange in patients with Guillain–Barré syndrome found that not only treatment with an IVIG dose of 0.4 g/kg body weight per day for 5 days was not only at least as effective as plasma exchange but also led to improved motor functions and hastened recovery in significantly more patients than plasma exchange. In addition, patients treated with IVIGs experienced fewer adverse events [108]. 

In a double-blind, multi-center trial in France, the optimal duration of IVIGs in Guillain–Barré syndrome was studied. The primary end-point was the time needed to regain the ability to walk with assistance. The study found that a longer course of 5 to 6 days of IVIG treatment (resulting in 2 g and 2.4 g total IVIG doses, respectively) leads to improved recovery, compared to a shorter, 3-day course (1.2 g total dose) [109].

Intravenous immunoglobulins have appeared to be crucial in managing acute exacerbations of neuromuscular disorders, most notably in Myasthenia Gravis and Lambert–Eaton Myasthenic syndrome. Myasthenia Gravis presents with fluctuating muscle weakness and pathological muscle fatiguability affecting the extraocular, bulbar, skeletal and respiratory muscles. The first trial comparing the effectiveness of IVIGs vs. plasma exchange in acute myasthenic crises was undertaken between 1996 and 2002 and found comparable results of both interventions with fewer adverse events and ease of application in the IVIG-treated group. Interestingly, a shorter 3-day course of 1.2 g total dose was superior to longer courses, contrasting with the findings in Guillain–Barré syndrome treatment [110].

IVIGs have also been implemented to manage Lambert–Eaton Myasthenic syndrome exacerbations. It is a disorder caused by autoantibodies directed against calcium voltage-gated membrane channels. In a placebo-controlled trial, a short course of 1 g/kg of IVIGs for 2 days markedly improved muscle strength and reduced serum calcium channel autoantibodies titers [111]. 

IVIGs also showed beneficial effects for Sjögren’s syndrome with severe neuropathy and limb weakness. IVIG administration improved all symptoms temporarily, but the long-term therapeutic benefit was not attained since symptoms resurfaced and worsened over time [112].

### 3.9. Other Autoimmune Diseases

However, many other autoimmune and immune-mediated conditions may benefit from IVIG administration. For the selected dermatological autoimmune disease (pemphigus vulgaris, pemphigus foliaceous, bullous pemphigoid, mucous membrane pemphigoid, epidermolysis bullosa acquisita, and cutaneous lupus erythematosus), IVIGs are commonly used, but as a second- or third-line treatment. Serious side effects were rare; the most common adverse effects reported were febrile infusion reactions, nausea, headache and fatigue [113]. 

A systemic review of Gao et al. on IVIG administration in livedoid vasculopathy (LV) concluded that IVIGs at a 1–2.1 g/kg body weight every 4 weeks is a safe and effective treatment alternative for refractory LV patients [114]. The patients demonstrated a good clinical response (i.e., reduction in pain, skin ulcerations and neurological symptoms) and decreased dependence on GCs and immunosuppressive agents. Moreover, IVIG infusions were well tolerated, and no severe adverse events were observed.

The studies on IVIG administration in patients with autoimmune diseases are presented in Table 1.

## 4. IVIGs as Immunomodulators in Patients with Reproduction Failures

### 4.1. IVIGs for Immune Cells Modulation

Different maternal immune cells and factors are involved in the immune tolerance toward the semi-allogeneic fetus during pregnancy [115]. Th1, Th2, Th17 and Treg cells are needed for embryo implantation and pregnancy maintenance. In decidua, NK cells also play an important role in cytokine generation, angiogenesis, vascular remodeling and trophoblast invasion [116,117]. 

Therefore, maternal–fetal immunological dysregulation could lead to infertility and embryo rejection. Immunological factors, such as anti-phospholipid antibodies or other autoantibodies, the increased level and cytotoxicity of NK cells in an intrauterine environment, and the increased ratios of Th1/Th2 and Th17/Treg cells and their related cytokine production may cause immunological attacks to the fetus, pregnancy loss or implantation failure [118]. Fostering immunological tolerance and reducing immunological rejection are essential for embryo protection and avoiding immune attacks. So, it seems that immunomodulatory and immunosuppressive drugs may control reproductive failures by controlling immune cells. There is growing research of evidence supporting the hypothesis that immunotherapy may improve birth rates and other positive outcomes for pregnant women [119]. 

The two main categories of immunotherapies in case of reproduction failure are “active” and “passive”. The injection of the father’s leukocytes triggers an immunological response in the mother, making the process active. On the other hand, infusions of IVIGs are used in passive immunotherapy to effect changes in the immune system. In the past, therapeutic strategies for miscarriage included prednisolone, intralipid and IVIGs. Nowadays, anti-TNF medications (Etanercept and Adalimumab) have been launched following these approaches to lessen adverse effects. However, IVIGs are still considered the gold standard for treatment. Many mechanisms mediate the immunomodulatory effects of IVIGs in the case of reproduction failure [120]. IVIGs protect the fetus by reducing complement deposition, activating the proliferation of suppressor T cells, down-regulating and up-regulating activator and inhibitory Fc cell surface receptors, and neutralizing anti-HLA antibodies. Furthermore, IVIG application may lessen the adherence of T cells to the primary constituents of the human placental extracellular matrix [121,122]. The mechanisms of IVIGs for reproductive failure are summarized in Figure 2.

IVIG-treated recurrent pregnancy losses (RPL) patients had lower Th1/Th2 and Th17/Treg cell ratios than untreated patients. IVIGs reduced the Th17/Treg cell ratio in low-quartile women [123,124,125]. 

In their extensive study and meta-analysis, Li et al. evaluated the effect of IVIGs on women undergoing in vitro fertilization (IVF) for infertility and/or early pregnancy loss. They found that IVIGs substantially impacted improvements in human fertility after analyzing the data from 8207 participants. The study results demonstrated a correlation between IVIG administration and increased implantations, clinical pregnancies and live births [126]. Later on, Ahmadi et al. reported that patients with RPL and immune cell abnormalities who receive IVIGs have a higher chance of achieving a successful pregnancy. It may cause a change in the Th1/Th2 balance, shifting the response toward the Th2, increasing Tregs and decreasing Th17 responses. However, patients with RIF, especially those with immunological abnormalities, may benefit from IVIGs to enhance implantation and pregnancy outcomes [121,122,123,124,125,126,127,128]. 

Moreover, there is also the possibility that IVIGs may enhance pregnancy outcomes by decreasing the cytotoxicity of NK cells and enhancing the number of NK inhibitory receptors. Through IVIGs, CD200 suppresses NK cytolytic activity [129,130]. 

Several studies indicate that treating individuals with reproductive failure with IVIGs, prednisone or a TNF inhibitor enhances the incidence of live births. Females with RPL and immunological etiologies, such as NK cell disease, had a better success rate when treated before conception than after that. IVIGs may protect the fetus from the mother’s immune system through many mechanisms, making it an effective treatment for improving pregnancy outcomes and live birth rates in patients with RM or RIF [131,132,133,134].

On the other hand, there are embryonic aneuploidies and their effect on pregnancy outcomes cannot be ignored even though all the scientific evidence sheds light on the role of IVIGs in RIF and RPL of immunological etiologies [135]. 

In conclusion, these therapies include manipulating immune cells to treat or prevent reproductive failure. Unfortunately, there are scarce data regarding treatment that can effectively and safely shield the developing baby from immune system assaults. However, further clinical trials and laboratory investigations are necessary to assess reproductive failure treatments.

### 4.2. Obstetric APS

The most substantial effects of APS seem to occur in pregnancy when they lead to a pregnancy loss rate as high as 80–90% [136,137]. Furthermore, obstetric APS should be managed by an experienced multidisciplinary team of specialists. One of the biggest challenges is that the optimal therapeutic target of APS treatment is elusive. Therefore, IVIGs were considered an emerging therapy option. Some case reports/series and observational or randomized trials confirmed the beneficial effects of IVIGs during pregnancy. However, the results are controversial. As shown above, IVIGs are the most commonly used in obstetric APS, where the administration of IVIGs aims to prevent obstetric complications, mainly recurrent pregnancy loss. 

Valensise et al. reported treating 14 women with APS with a history of recurrent spontaneous abortions with 0.5 g/kg IVIGs for 2 consecutive days from the 5 gestational weeks of pregnancy and repeated every 4 weeks until the 33rd week, with excellent safety profile [138].

Similarly, Clark et al. [139] reported a live-birth rate of 84% in 15 women with APS treated with monthly IVIGs along with LDA, sub-cutaneous heparin and steroid. Moreover, the authors reported a significant decrease in aCL antibody levels in seven pregnancies. Watanabe et al. [140] also confirmed the benefit of IVIG therapy (a 5-day course at 0.4 mg/kg from 6–7 gestational weeks) in APS pregnant women, in combination with unfractionated heparin injection, LDA, and prednisone. No bleeding or thrombotic incidents were demonstrated, and all three patients achieved live births.

However, most studies on IVIGs during pregnancy are observational and not often randomized trials. The pilot study for IVIG treatment in APS women with APS and 3 or more consecutive spontaneous first trimester abortions reported that 0.3 mg/kg IVIGs were administered as soon as the pregnancy was registered, at 3 weekly intervals until the 16–17th weeks of gestation, resulting in 31/34 successful pregnancies (from that point continued beyond the first trimester) [141]. In 1998, Sher et al. [142] demonstrated that IVIGs improved birthrates in aPL-positive women during IVF (treated in combination with heparin and aspirin). A randomized, double-blind, placebo-controlled pilot trial by Branch et al. [143] reported excellent obstetric outcomes (i.e., delivering live-born infants). Moreover, IVIGs did not reduce the rate of obstetric or neonatal complications. 

Subsequently, Vaquero et al., in their prospective two-centers trial [144], reported fewer cases of gestational hypertension and gestational diabetes in the IVIG-treated group compared to prednisone-plus LDA-treated patients, with equivalent live-birth rates. A study conducted in 2002 [145] showed favorable fetal outcomes in APS women treated with IVIGs in combination with LDA and heparin, who were non-responders to the standard regimen. Other studies presented contrasting results [146,147]—more live births in the group of APS pregnant women treated with standard anticoagulation therapy or no significant differences between women treated with IVIGs and women on standard treatment [148]. The other two papers show promising results and the superiority of comprehensive treatment (including IVIGs) in APS women to decrease pregnancy complications, although without differences in the abortion rate [149,150].

In sum, at the moment, IVIG therapy for obstetric APS should be reserved for selected patients (i.e., resistant to conventional treatment, presence of other autoimmune conditions, infections, in case of contraindicated anticoagulation) [70].

## 5. Conclusions

The immunoregulatory effects of IVIGs in autoimmune and inflammatory diseases depend on various mechanisms, some of which are still elusive. IVIG immunomodulatory potential in patients with different immune-mediated, inflammatory and autoimmune diseased results from a number of complex mechanisms working together when administered as immunomodulators. It is assumed that IVIGs influence more than one immune arm, with many innate and adaptive immune pathways being targeted. In addition, many non-immunological but not mutually exclusive effects have been demonstrated. Therefore, it is difficult to determine a common mechanistic understanding of the IVIG mode of action.

The studies available, including case reports, case series, observational studies, randomized controlled trials, systematic reviews and meta-analyses, confirmed IVIG administration benefits in efficacy and safety in patients with selected autoimmune, immune-mediated, inflammatory conditions, as well as in reproduction failure cases.

## Figures and Tables

**Figure 1 antibodies-12-00020-f001:**
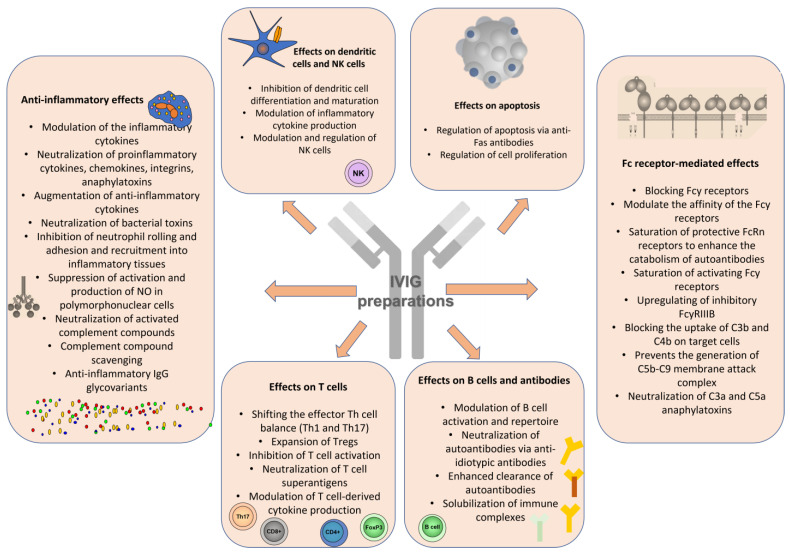
Immune mechanisms exerted by IVIG therapy.

**Figure 2 antibodies-12-00020-f002:**
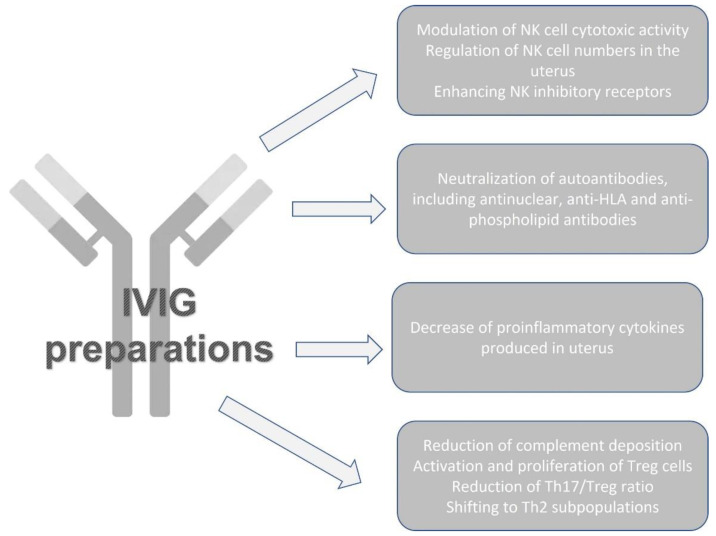
Effects of IVIG therapy on immune mechanisms associated with reproductive failure.

**Table 1 antibodies-12-00020-t001:** Studies available on patients with autoimmune diseases, treated with IVIGs.

Autoimmune Condition	Type of Study	Patients	Dosage Regimen	Results	Ref.
Idiopathic inflammatory myopathies	Double-blind, placebo-controlled	15 biopsy-proved, treatment-resistant dermatomyositis	2 g per kilogram of body weight or placebo per month for 3 months	Significant improvement in scores of muscle strength and neuromuscular symptoms	[37]
	Retrospective	Steroid-refractory esophageal involvement related to polymyositis and dermatomyositis	2 mg/kg monthly	82.6% of patients exhibited resolution of esophageal impairment	[38]
	Open	35 patients with chronic, refractory polymyositis	1 mg/kg/day for 2 consecutive days per month	Significant clinical improvement in 71.4% and biochemical response	[40]
	Single-center long-term follow-up	91 patients with polymyositis and dermatomyositis	1 g/kg (5 g/h) on two consecutive days each month for six months	Better survival in IVIG/subcutaneous Ig treated patients	[42]
	Prospective, double-blind, randomized, placebo-controlled phase III study-ProDERM Study	Patients with dermatomiositis	2 g/kg of IVIg or placebo, every 4 weeks until week 16	Improvement in total improvement score, time to improvement, proportions of patients with deteriorations	[44]
	randomized, placebo-controlled, ProDERM	95 patients with dermatomiositis	2 g/kg of IVIg or placebo, every 4 weeks until week 16	79% had a total improvement score of at least 20; at least moderate improvement and major improvement	[45]
Systemic lupus eryhtmeatodes	Systematic review and meta-analysis (3 controlled and 10 observational studies)	Heterogenous group of SLE patients, subjects with lupus nephritis, hematological and cutaneous involvement	400 mg/kg/d over 5 days	Response rate of 30.9%; reduction of SLE disease score	[4]
	Pilot	12 patients with mildly to moderately active SLE	30 g of sulfonated IVIG preparation on each of Days 1–4 and 21–24	Systemic Lupus Activity Measure dropped significantly, lasted 5–12 months, decline in anti-dsDNA antibodies	[57]
	Observational	20 SLE patients	2 g/kg IVIg monthly, in a 5-d schedule	Beneficial clinical response, more responsive to treatment-arthritis, fever, thrombocytopenia, and neuropsychiatric lupus	[60]
Anti-phospholipid syndrome	Review of the literature	Patients with obstetric APS	0.4–1 g/kg/month	Prevent recurrent thrombosis in APS patients refractory to conventional anticoagulant treatment	[70]
Systemic sclerosis	Preliminary report	3 patients with systemic sclerosis	2 g/kg at six courses	large decrease in the skin score, no changes in anti-PM-Scl antibodies	[77]
	Pilot	7 women with systemic sclerosis	2 g/kg body weight during 4 days/month for six consecutive courses	Decrease in joint pain and tenderness, hand function improved together with quality of life, the skin score reduced	[78]
	Retrospective	46 patients with systemic sclerosis	at least 1 IVIG infusion at a dosage > 1 g/kg/cycle	Significant improvement of muscle pain, muscle weakness, joint pain, CK and CRP levels	[82]
Kawasaki disease	Systematic review and meta-analysis	14 studies with 70,396 patients	High dose	Reduced risk of coronary artery lesions; Early treatment with IVIG can lead to an increased risk of IVIG unresponsiveness	[90]
ANCA-associated vasculitides	Randomized, controlled	34 patients with Wegener granulomatosis	2 g/kg	Fall in disease activity score	[94]
	French Nationwide	92 patients with granulomatosis with polyangiitis (Wegener’s), eosinophilic granulomatosis with polyangiitis (Churg–Strauss), or microscopic polyangiitis	1 mg/kg/d for 2 days, 0.5 mg/kg/d for 4 days, other	Remission in 56% of patients at 6th month	[97]
Autoimmune dysautonomias	Retrospective	38 patients with disabling, refractory autoimmune dysautonomias, incl. postural tachycardia syndrome and gastrointestinal dysmotility	1 mg/kg monthly for 3 months	Improved composite autonomic symptom scale and functional/ability score in 83.5% of patients	[101]
Autoimmune neurological diseases	Meta-analysis	23 reports with 344 patients with Guillain–Barre, autoimmune encephalitis, etc.	1–2 g/kg at an average	Beneficial effect of IVIG administration on patient improvement over placebo and identical effects to plasmapheresis	[104]
Guillain–Barré syndrome	Multi-center, randomized	150 patients with Guillain–Barré syndrome	5 doses of 0.4 g per kilogram per day	Improved strength, significantly fewer complications and less need for artificial ventilation	[108]
	Single-center trial	36 patients with Guillain–Barré syndrome with severe hemostasis, unstable hemodynamics, or uncontrolled sepsis	0.4 g/kg/day IVIg for 3 or 6 days	Improvement in walking without assistance and needed for ventilation	[109]
Myasthenia gravis	Randomized clinical trial	87 patients with myasthenia gravis	0.4 mg/kg daily	Improvement of myasthenic muscular score	[110]
Livedoid vasculopathy	Systematic review	17 articles-80 patients with livedoid vasculopathy	1–2.1 g/kg body weight every 4 weeks	Resolution of pain, skin ulcerations, and neurological symptoms, and reducing the dependence on glucocorticoids and immunosuppressive agents	[114]
Sjögren syndrome with severe neuropathy and limb weakness	Cross-sectional	184 patients with Neuro-Sjögren	N/A	Improvement in motor function or stabilization of status quo; temporary improvement of all symptoms, but long-term clinical benefit could not be achieved as symptoms relapsed	[112]
Dermatological autoimmune diseases	Retrospective	Pemphigus vulgaris, pemphigus foliaceous, bullous pemphigoid, mucous membrane pemphigoid, epidermolysis bullosa acquisita, and cutaneous lupus erythematosus	2 g per kg body weight distributed over 2–5 days every 4 weeks	Achieved diseases control, reduction of immunosuppressive therapy	[113]

## Data Availability

No new data were created or analyzed in this study. Data sharing is not applicable to this article.

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
