# Peer review of "Intravenous Immunoglobulins as Immunomodulators in Autoimmune Diseases and Reproductive Medicine"

_2073-4468, 2023, doi:10.3390/antib12010020_

Round 1
Reviewer 1 Report
The authors provide a well-written review paper on the use of IVIG in autoimmune diseases and reproductive medicine. I have, however, several points that need to be addressed/changed.
Major points:
1. Suggest changing figure 1. While the provided overview is helpful, it appears a bit confusing. A different way of providing this information would be to cluster the proposed mechanism according to cell type/cytokine.
2. page 3, lines 86-98: there are very few references to support the statements made in this paragraph. Only reference 10 is cited twice. The authors need to provide more evidence from original papers to support their claims here.
3. The whole section 2 "Immune mechanisms..." would benefit from more subheadings to guide the reader. Suggest using subheadings according to suggested IVIG effects (e. g. NKc ells.. T-cells, FCr, etc.)
4. For section 3 "IVIGs treatment for autoimmune disease", I suggest the addition of a table with an overview of only RCTs of IVIGs in the various conditions. This would be very helpful for the readers.
5. paragraph 3.1 on myositis contains errors and is incomplete: Please add the current classification of myositis (overlap-myositis, immune-mediated necrotizing .., antisynthetase-syndrome, dermatomyositis, and polymyositis), because IVIG play a role also in these (especially IMNM!).
6. the ProDERM study is entirely missing! (DOI: 10.1097/MD.0000000000023677) Please add and discuss.
7. The section on SLE is unnecessarily long. Apart from hematological manifestations, IVIG are not very frequently used. In LN, I dont think that IVIG has any use in clinical practice. This paragraph can be condensed. Lines 243-259 are missing references.
8. section 3.8 on autoimmune neurological disorders should include a paragraph on Neuro-Sjögren's, where IVIG are also used. Ref example: Neuro-Sjögren: Peripheral Neuropathy With Limb Weakness in Sjögren's Syndrome.
Front Immunol. 2019 Jul 11;10:1600. doi: 10.3389/fimmu.2019.01600. eCollection 2019.
Minor points:
1. page 4, lines 136-138: the sentence appears incomplete.
2. page 7 line 288: "neurolupus". The commonly used term is "neuropsychiatric SLE, NPSLE).
3. section 3.3 some comments on the use of IVIG in CAPS (catastrophic APS) are missing
4. Rename section 3.4 to simply "Systemic sclerosis"
5. section 3.5: a dose for IVIG in Kawasaki should nevertheless be provided
6. section 3.6 AAV: typo in line 386 (AASV); the authors should add the current ACR guidelines; page 9 line 402: a sentence fragment appears to be missing here.
7. I think that dermatomyositis, systemic vasculitis, and SLE should be removed from section 3.9
8. Title of section 4 should be changed since the authors also mention obstetric APS here. A figure could be considered if deemed appropriate by the authors illustrating the mechanisms in ARTs. page 11, line 510 ("thesis") change to "hypothesis"
9. page 11 lines 493-495 contain sentences from the template that should be deleted.
10. page 12, lines 579-593: inappropriate use of hyphens ("-") that should be changed to ":" or ";", as appropriate.
11. For section 4.1. and 4.2, the conclusions should be clearer formulated.
Author Response
Dear Editor,
Dear reviewers,
Thank you for your time in reviewing our paper. We acknowledge that our paper might have some issues in conformity with the reviewers` comments. Therefore, we have revised the manuscript extensively using track changes/highlighting and point-to-point responses to all the recommendations.
Round 1
Reviewer 1
The authors provide a well-written review paper on the use of IVIG in autoimmune diseases and reproductive medicine. I have, however, several points that need to be addressed/changed.
- Thank you very much for the critical notes and the overall evaluation of our paper as good. We appreciate the opportunity to revise our article accordingly.
Major points:
- Suggest changing figure 1. While the provided overview is helpful, it appears a bit confusing. A different way of providing this information would be to cluster the proposed mechanism according to cell type/cytokine.
- We agree that the figure will benefit if presented differently. We revised it.
- page 3, lines 86-98: there are very few references to support the statements made in this paragraph. Only reference 10 is cited twice. The authors need to provide more evidence from original papers to support their claims here.
- We completely agree with the referee and added references supporting the paragraph's statements.
- The whole section 2 "Immune mechanisms..." would benefit from more subheadings to guide the reader. Suggest using subheadings according to suggested IVIG effects (e. g. NKc ells.. T-cells, FCr, etc.)
- Thank you for the valuable comment. We did our best to improve the clarity by dividing the section into subsections.
- For section 3 "IVIGs treatment for autoimmune disease", I suggest the addition of a table with an overview of only RCTs of IVIGs in the various conditions. This would be very helpful for the readers.
- Thank you very much for the suggestion. We found it particularly valuable. But unfortunately, at this point, we did not prepare a table with all the studies on IVIG treatment for various conditions because we included almost all indications, and such a table would be tremendously huge.
- However, we are ready to prepare such if the referee insists. We believe that this will be very valuable for the readers.
- Furthermore, as the referee suggested, we revised figure 1 and added an additional figure for IVIGs in reproduction medicine. We believe this would benefit the quality of the paper as well.
- paragraph 3.1 on myositis contains errors and is incomplete: Please add the current classification of myositis (overlap-myositis, immune-mediated necrotizing .., antisynthetase-syndrome, dermatomyositis, and polymyositis) because IVIG play a role also in these (especially IMNM!).
- Thank you for the valuable point. We supplement the classification of myopathies. We also added a passage on the IVIG for IMNM.
- the ProDERM study is entirely missing! (DOI: 10.1097/MD.0000000000023677) Please add and discuss.
- Thank you for this comment. We added two papers related to the ProDERM study and discussed the results.
- The section on SLE is unnecessarily long. Apart from hematological manifestations, IVIG are not very frequently used. In LN, I dont think that IVIG has any use in clinical practice. This paragraph can be condensed. Lines 243-259 are missing references.
- We agree with the referee that the FDA has not approved IVIG treatment for SLE and neuro-lupus/neuropsychiatric SLE. We tried to shorten the section but eventually left the details because IVIGs` mechanisms in SLE are well-documented, and we can rely on these data.
- section 3.8 on autoimmune neurological disorders should include a paragraph on Neuro-Sjögren's, where IVIG are also used. Ref example: Neuro-Sjögren: Peripheral Neuropathy With Limb Weakness in Sjögren's Syndrome.
Seeliger T, Prenzler NK, Gingele S, Seeliger B, Körner S, Thiele T, Bönig L, Sühs KW, Witte T, Stangel M, Skripuletz T. Front Immunol. 2019 Jul 11;10:1600. doi: 10.3389/fimmu.2019.01600. eCollection 2019.
- We also appreciate this comment, and we added this information.
Minor points:
- page 4, lines 136-138: the sentence appears incomplete.
- page 7 line 288: "neurolupus". The commonly used term is "neuropsychiatric SLE, NPSLE).
- Thank you very much. We corrected the issue.
- section 3.3 some comments on the use of IVIG in CAPS (catastrophic APS) are missing
- We added information on this topic.
- Rename section 3.4 to simply "Systemic sclerosis"
- We agreed and corrected the issue.
- section 3.5: a dose for IVIG in Kawasaki should nevertheless be provided
- section 3.6 AAV: typo in line 386 (AASV); the authors should add the current ACR guidelines; page 9 line 402: a sentence fragment appears to be missing here.
- Thank you. All the issues are resolved.
- I think that dermatomyositis, systemic vasculitis, and SLE should be removed from section 3.9
- We agree. Done.
- Title of section 4 should be changed since the authors also mention obstetric APS here. A figure could be considered if deemed appropriate by the authors illustrating the mechanisms in ARTs. page 11, line 510 ("thesis") change to "hypothesis"
- Thank you for the great suggestions. We considered and applied them all.
- page 11 lines 493-495 contain sentences from the template that should be deleted.
- Thank you for noticing that. We corrected the issue.
- page 12, lines 579-593: inappropriate use of hyphens ("-") that should be changed to ":" or ";", as appropriate.
- For section 4.1. and 4.2, the conclusions should be clearer formulated.
- We revised the conclusions of these subsections to make them clearer and more concise.
Reviewer 2 Report
The review discusses intravenous immunoglobulins as a therapy in autoimmune diseases and reproductive medicine.
IVIG is interchangeably being used as a singular and a plural subject of a sentence. Please check the subject-verb agreement throughout the manuscript and keep it consistent (For example, see lines 67-68). In addition, there are several instances in which the sentences are very long and may need to be shortened to improve ease of reading and understanding.
Line 34: “Y shaped protein” is not really a technical description of an immunoglobulin. Please consider revising.
Figure 1 would be more informative if it was presented as a table with references for each mechanism. In addition, I suggest the addition of a table that has each disease along with the treatment regimes tested, endpoints to test efficacy, and references. Finally, I suggest the addition of figures that show mechanisms whereby IVIG therapy can mediate pro- or anti-inflammatory effects.
Line 178: I suggest adding a sentence that describes what will be discussed in the next sections. This will help for organizational purposes.
What is known about IVIG therapy in RA and in hematological disorders. These are discussed in the introductory section, but are not further discussed.
Line 215: reference is in the incorrect format.
Lines 493-495: I think this text is not meant to be here.
Author Response
Dear Editor,
Dear reviewers,
Thank you for your time in reviewing our paper. We acknowledge that our paper might have some issues in conformity with the reviewers` comments. Therefore, we have revised the manuscript extensively using track changes/highlighting and point-to-point responses to all the recommendations.
The review discusses intravenous immunoglobulins as a therapy in autoimmune diseases and reproductive medicine.
IVIG is interchangeably being used as a singular and a plural subject of a sentence. Please check the subject-verb agreement throughout the manuscript and keep it consistent (For example, see lines 67-68). In addition, there are several instances in which the sentences are very long and may need to be shortened to improve ease of reading and understanding.
- Thank you for the critical notes and the opportunity to revise our paper accordingly.
- We corrected the issue with subject-verb agreement and used IVIGs as subject.
- We did our best to improve the clarity by making the language plain and concise.
Line 34: “Y shaped protein” is not really a technical description of an immunoglobulin. Please consider revising.
- Thank you for the critical note. We corrected the issue.
Figure 1 would be more informative if it was presented as a table with references for each mechanism. In addition, I suggest the addition of a table that has each disease along with the treatment regimes tested, endpoints to test efficacy, and references. Finally, I suggest the addition of figures that show mechanisms whereby IVIG therapy can mediate pro- or anti-inflammatory effects.
- Furthermore, we revised figure 1 – with particular attention to their immunological effects and added an additional figure for IVIG in reproduction, as the referee suggested. We believe this would benefit the quality of the paper as well.
- Thank you very much for suggesting the table with all the indications and diseases. We found it particularly valuable. But unfortunately, at this point, we did not prepare a table with all the studies on IVIG treatment for various conditions because we included almost all indications, and such a table would be tremendously huge.
- However, if the referee insists, we are ready to prepare such a table. We believe this will be very valuable for the readers and will prepare such a table, regardless of how extensive it would be.
Line 178: I suggest adding a sentence that describes what will be discussed in the next sections. This will help for organizational purposes.
- Thank you for the great suggestion. We added this in the last paragraph.
What is known about IVIG therapy in RA and in hematological disorders. These are discussed in the introductory section, but are not further discussed.
- The referee is right to point out that we did not cover these conditions in the paper. However, since these conditions have a substantial clinical impact, we add a couple of sentences in section 3.
- This is outside our scope regarding hematological diseases, and we only enlisted these disorders as indicated for IVIG therapy.
Line 215: reference is in the incorrect format.
- Thank you for noticing this. We missed adding this reference in the first submitted manuscript, and now we corrected the issue.
Lines 493-495: I think this text is not meant to be here.
- Thank you for noticing that. We corrected the issue.
Round 2
Reviewer 1 Report
The authors have improved and followed the recommendations. However, I really do recommend shortening section 3, as IVIg are not equally important across all diseases mentioned.
Further, I recommend (as previously suggested) the addition of a table with a detailed overview of all available trials for IVIg in autoimmune disease. I acknowledge that this is extra work, but it would really improve the paper a lot.
Author Response
- Thank you once again for your dedicated time to providing us with these beneficial suggestions and critics and for the opportunity to revise our manuscript further.
- We agree with the referee that section 3 is relatively too big compared to the other sections. Therefore, we reviewed section 3 to reduce the material. However, our attempts to shorten the section were unsuccessful since the logic flow had been interrupted, and some of the facts are important for the following sections. Therefore, we would be grateful if the referee approved the final version.
- We added a table on the autoimmune disease, where IVIG treatment was performed, to summarize the data and make the paper a more comprehensive source of information.
Reviewer 2 Report
I still think there are some unresolved issues with the subject verb agreement. Please check other published manuscripts to determine the convention of using IVIG vs. IVIGs, and whether it is singular or plural.
Otherwise, thank you for addressing my comments.
Author Response
- Thank you once again for your dedicated time to providing us with these beneficial suggestions and critics and for the opportunity to revise our manuscript further.
- We spotted these problematic parts and revised them accordingly.